# Synthesis and Evaluation of AlgNa-g-Poly(QCL-co-HEMA) Hydrogels as Platform for Chondrocyte Proliferation and Controlled Release of Betamethasone

**DOI:** 10.3390/ijms22115730

**Published:** 2021-05-27

**Authors:** Jomarien García-Couce, Marioly Vernhes, Nancy Bada, Lissette Agüero, Oscar Valdés, José Alvarez-Barreto, Gastón Fuentes, Amisel Almirall, Luis J. Cruz

**Affiliations:** 1Centro de Biomateriales, Universidad de La Habana, La Habana 10400, Cuba; jgcouce@gmail.com (J.G.-C.); bada@biomat.uh.cu (N.B.); lissetteaguerol@gmail.com (L.A.); amisel.almirall@gmail.com (A.A.); 2TNI Group, Department of Radiology, Leiden University Medical Center (LUMC), 2333 ZA Leiden, The Netherlands; L.J.Cruz_Ricondo@lumc.nl; 3Laboratorio de Biología Molecular, Departamento de Radiobiología, Centro de Aplicaciones Tecnológicas y Desarrollo Nuclear (CEADEN), La Habana 11300, Cuba; mariolys@ceaden.edu.cu; 4Centro de Investigación de Estudios Avanzados del Maule (CIEAM), Vicerrectoría de Investigación y Posgrado, Universidad Católica del Maule, Talca 3460000, Chile; ovaldes@ucm.cl; 5Department of Chemical Engineering, Universidad de San Francisco de Quito, Quito 170901, Ecuador; jalvarezb@usfq.edu.ec

**Keywords:** hydrogels, sodium alginate, betamethasone, drug delivery, cartilage tissue engineering

## Abstract

Hydrogels obtained from combining different polymers are an interesting strategy for developing controlled release system platforms and tissue engineering scaffolds. In this study, the applicability of sodium alginate-g-(QCL-co-HEMA) hydrogels for these biomedical applications was evaluated. Hydrogels were synthesized by free-radical polymerization using a different concentration of the components. The hydrogels were characterized by Fourier transform-infrared spectroscopy, scanning electron microscopy, and a swelling degree. Betamethasone release as well as the in vitro cytocompatibility with chondrocytes and fibroblast cells were also evaluated. Scanning electron microscopy confirmed the porous surface morphology of the hydrogels in all cases. The swelling percent was determined at a different pH and was observed to be pH-sensitive. The controlled release behavior of betamethasone from the matrices was investigated in PBS media (pH = 7.4) and the drug was released in a controlled manner for up to 8 h. Human chondrocytes and fibroblasts were cultured on the hydrogels. The MTS assay showed that almost all hydrogels are cytocompatibles and an increase of proliferation in both cell types after one week of incubation was observed by the Live/Dead^®^ assay. These results demonstrate that these hydrogels are attractive materials for pharmaceutical and biomedical applications due to their characteristics, their release kinetics, and biocompatibility.

## 1. Introduction

Articular cartilage is a highly specific, avascular, connective tissue that lines the end of each bone and forms the joint, providing a smooth and gliding surface. It is predominantly formed of water, proteoglycans, and type II collagen. Chondrocytes are the only functional cells that compose articular cartilage [1,2]. Osteoarthritis (OA) is the most common condition that impacts cartilage. This is a disease characterized by the loss of the matrix and its functionality, producing inflammation and severe pain, which are symptoms usually treated with anti-inflammatory drugs. Corticosteroids (e.g., dexamethasone, triamcinolone, and betamethasone) are frequently used to treat OA due to their potent anti-inflammatory activity. It has been proven that a low dose exerts effects on down-regulating the gene expression level of MMPs (MMP-1, MMP-3, and MMP-13) in chondrocytes, reducing glycosaminoglycans loss [1]. However, their prolonged administration causes many side effects. In more advanced stages of the disease and due to the limited capacity for self-repair, replacement or induced regeneration of the damaged area is often required [3]. To date, different treatment methods have been designed, such as mosaicplasty, osteochondral allograft transplantation, and autologous chondrocyte implantation, among others. However, there are still limitations to producing cartilage with full biological activity [2,4]. Therefore, tissue engineering continues to be an interesting and promising alternative to repair cartilage defects in a functional way and to apply more localized treatments to reduce side effects [5,6].

Hydrogels are three-dimensional polymeric structures, capable of retaining large amounts of water or fluids without dissolving or losing their integrity. Due to these characteristics, they have a high similarity to living tissues, and they are suitable platforms for tissue engineering, drug-carrying matrices, artificial articular cartilage, and smart devices to external stimuli, such as pH and temperature [7]. To treat joint cartilage damage, hydrogels are created to be resident substitutes when replacing injured cartilage or to be used as cell-laden or drug-laden components that promote or stimulate tissue regeneration [8]. The use of natural polymers, such as polysaccharides, for developing these matrices has been frequently studied in past decades. Chitosan, hyaluronic acid, and sodium alginate are among the most widely used [3]. Sodium alginate (AlgNa) is a linear anionic polysaccharide, composed of β-D mannuronic acid and α-L-guluronic acid units, which are obtained mainly from brown marine algae [9,10]. Due to its properties (biocompatible, biodegradable, and non-toxic), several materials have been developed from alginate, such as microparticles, hydrogels, and films, for applications in the biomedical field [11]. Specifically, sodium alginate hydrogels have a similar structure to natural extracellular matrix, and, for that reason, their use in tissue engineering and drug delivery has grown exponentially [11,12], which previous studies carried out by different authors have demonstrated. Yamaoka et al. showed that the alginate matrices studied help chondrocytes reduce cell-to-cell contacts and maintain cell shape and function. These properties can improve the expression and accumulation of cartilaginous matrices, such as type II collagen and GAG [13]. In another work, Baghaban Eslaminejad et al. [14] observed the presence of abundant microvilli developed on the surfaces of chondrocytes seeded in alginate matrices, which is indicative of cellular participation in the secretion of the matrix by scanning electron microscopy. This morphology, in turn, would be the result of the active interaction of chondrocytes with alginate molecules. In another study, the incorporation of alginate in poly(2-hydroxyethyl methacrylate) hydrogels increased the amount of glycosaminoglycans and the proliferation of cultured chondrocytes [7]. However, hydrogels obtained only from natural polymers are mechanically weak due to the large amount of water they absorb. As a solution to this limitation, synthetic polymers are usually added to improve the physicochemical properties of hydrogels. Among the synthetic polymers, poly(2-hydroxyethyl methacrylate) (HEMA), which is a neutral hydrophilic polymer, has been one of the most used in tissue engineering due to its versatile properties, such as biocompatibility, good mechanical properties, and ease of synthesis [4,5,15]. Hydrogels for cartilage tissue engineering from the combination of polymers, natural (e.g., chitosan, collagen), and synthetic (polyethylene glycol and polyvinyl alcohol) have been reported by several authors [4,11,16]. 2-acryloyloxyethyl trimethyl ammonium chloride (QCL) is a positively charged monomer composed of a polymerizable double bond and a quaternary ammonium at the end of its structure. According to previous reports, compounds with quaternary ammonium groups in their structure have been used as vectors to target therapeutic agents (anti-inflammatory and osteoarthritis modifying drugs) toward the cartilage, and to interact with sulfate and carboxyl groups present in the proteoglycans of ECM [17,18,19,20]. It is known that QCL’s homopolymers and copolymers are highly charged structures, capable of interacting through electrostatic attractions with other oppositely charged compounds [19,20,21,22]. The inclusion of hydrogels designed for cartilage tissue engineering provides an additional benefit because they improve the tissue-material interaction.

In this work, AlgNa-g-poly(QCL-co-HEMA) hydrogels are prepared by radical copolymerization of HEMA and QCL in the presence of AlgNa and *N*,*N*-methylene diacrylamide (MBA) as a crosslinking agent. Although there are some similar reports on systems based on one or more of these components [10,23,24,25,26], this specific composition has not been reported or studied as a drug carrier and platform for chondrocytes proliferation. The structure and morphology are analyzed by Fourier transform-infrared spectroscopy (FTIR) and scanning electron microscopy (SEM). The swelling capacity is measured as a function of time and pH. The betamethasone (BTM) release from the prepared matrices in phosphate buffered saline (PBS) is evaluated. Furthermore, the cytocompatibility in vitro with C-28 and 3T3 cells using MTS and Live/Dead^®^ assays is evaluated. All analyses are carried out based on the composition of the hydrogels.

## 2. Results and Discussion

### 2.1. AlgNa-g-Poly(QCL-co-HEMA) Hydrogels

The hydrogels were obtained by a mechanism of copolymerization and simultaneous chemical crosslinking, as shown in Figure 1. Initially, the thermal decomposition of the potassium persulfate (KPS, initiator) occurs, generating the anionic sulfate radicals. Next, the radicals interact with the -OH group of AlgNa, extracting the hydrogen to form the corresponding macroradicals. These macroradicals act as active centers through which the grafting of QCL and HEMA monomer molecules in the main chains of AlgNa begins. Chains propagation and a simultaneous cross-linking process with MBA occur, forming the three-dimensional graft copolymer AlgNa-g-poly(QCL-co-HEMA). Mechanisms similar to the one proposed were reported in previous works and in other systems [24,25,26,27].

### 2.2. Hydrogel Characterizations

#### 2.2.1. FTIR Spectroscopy Analysis

FT-IR spectroscopy was used to confirm the structure of the prepared hydrogels, the spectrums of the initial pure components, and some samples of hydrogels, which are shown in Figure 2. Figure 2a shows the spectrum of AlgNa. The broad band at 3258 cm^−1^ can be attributed to an -OH stretching vibration. Other characteristic signals observed at 1598 and 1403 cm^−1^ are attributed to asymmetric and symmetric stretching vibration bands of the -COO^‾^ group [26]. Signals at 1080, 1025, and 813 cm^−1^ correspond to stretching vibrations of asymmetric and symmetric -C-O-C and Na-O, respectively [28]. Both monomers employed have C = O and C = C groups in their structure whose characteristic peaks are observed at 1718 and 1636 cm^−1^ in the QCL spectra (Figure 2b) [29,30] and at 1711 and 1636 cm^−1^ in the case of HEMA (Figure 2c) [31,32]. In the QCL spectrum, other characteristic peaks are also observed at 1478 and 1185 cm^−1^, which are attributed to the bending band of quaternary ammonium groups (-N^+^(CH_3_)_3_) and the C-O of ester, respectively [29,33]. At 950 cm^−1^, a singular band related to the C–N stretching of quaternary ammonium groups appears [34].

In addition to the C=O and C=C characteristic bands, HEMA shows intense signals at 940 and 813 cm^−1^ associated with the -C=C- group [35], and, at 1160 and 1076 cm^−1^, corresponding to the C-O-C stretching vibrations of the ester group [25,31]. In the hydrogel spectrums (Figure 2d–f), all absorption bands mentioned above are present while some of them overlap. The signals between 1720 and 1724 cm^−1^ are due to the C=O stretching frequency, while those between 1070 and 1185 cm^−1^ are caused by the stretching frequencies of C-O-C bonds of ester groups, demonstrating the presence of the QCL and HEMA units in the gel network. Additionally, the intensity of bands at 1478 and 950 cm^−1^, ascribed to QCL, increases between the spectrum 2d and 2f due to the growth of QCL in the hydrogel’s composition. Signals at 1598 cm^−1^ are due to the C=O of the carboxylate group of alginates, whereas the bands at 1025 cm^−1^ are due to the symmetric C-O-C stretching frequencies. In addition, the asymmetric stretching vibration of the COO^‾^ group of alginate at 1402 cm^−1^ remains nearly the same in the hydrogels, indicating a lack of chemical interactions occurs during the hydrogel’s obtention process [24]. On the other hand, the bands at 1636, 940, and 813 cm^−1^ associated with the C=C stretching vibration were not found in the spectrum of the samples H2 and H4 (Figure 2d,e), while the spectrum 2f shows a small band at 1636 cm^−1^. This result was only observed for sample H6, which contains the highest percentage of QCL. This is a charged monomer, and also has three methyl factors linked at the end of the chain, forming a bulky group. Both conditions can cause the approach between the monomer units to be limited, which implies that their inclusion in the growing polymer chains is reduced and, therefore, some unreacted monomers remained, causing the corresponding signal of methylene.

#### 2.2.2. Morphology Characterization

Figure 3 (H1 to H6) show the SEM micrographs of the internal morphology of AlgNa-g-poly (QCL-co-HEMA) hydrogels. It is observed that all the samples have a porous structure, and, although the pore size varies, it does not show a marked difference with respect to variation of the sample’s composition. In the SEM images, it is observed that hydrogels have interconnected pores through which the water molecules or fluids can easily spread and, therefore, act on the swelling degree and rate of the material, as well as constitute as sites of external stimuli interactions, such as pH or temperature. This porous structure is appropriate to allow attachment, growth, and proliferation of cells inside the matrix.

#### 2.2.3. Swelling Studies

The swelling process of hydrogels is conditioned by different factors, of which the matrix composition and pH of the medium have been extensively studied. The matrices developed in this research consist of three components of which each one contributes a functional group with hydrophilic characteristics (-COOH, -N^+^(CH_3_)_3_, -OH^‾^). Now, these groups can, in turn, undergo interactions with each other depending on the pH of the medium. Futhermore, the swelling process in the hydrogels under study will be strongly influenced by their composition and the pH of the medium. Figure 4A,B show the swelling profiles of the hydrogels at pH 2.0 and 7.4, respectively, and Figure 4C shows the maximum swelling at equilibrium in more detail.

The results show that there is a significant difference in the swelling of the hydrogel samples at different pH values. It can be seen that the swelling degree is higher at pH 2.0 than at pH 7.4 in all compositions, which is contrary to what is described in the literature for matrices containing AlgNa in their structure [36,37]. This result is due to the fact that AlgNa has a pKa between 3.2–4.0, so, at pH 7.4, the carboxylic groups of its units are in a carboxylates form (-COO^−^) and can be linked through electrostatic interactions with the N^+^(CH_3_)_3_ groups of QCL (Figure 5), causing the chains to shrink and the matrix to become more compact, reducing the fluid rate entry. At pH 2.0, the carboxylate ions are protonated, and the carboxyl group is in acid form. Hence, electrostatic interactions do not occur. In addition, it can be observed that, at pH 7.4, the swelling degree of the hydrogel changes depending on the composition. Samples containing 20% AlgNa in their composition have less swelling when compared with samples that have 10% AlgNa. These results reaffirm that electrostatic interactions caused by the AlgNa structure have a major role in the swelling process.

It is also valid to discuss the fact that, at pH 2.0, although electrostatic interactions do not occur, hydrogels show a similar swelling behavior as compared to the one observed at pH 7.4 regarding the composition of the materials. In this case, hydrogen bond formation among carboxylic groups with each other and with OH^−^ groups of the sugar moiety of the AlgNa structure are the predominant interactions produced, which act as a barrier that strengthens the polymeric network and hinders the entrance of water molecules inside the hydrogel network [36,38]. According to this, the samples containing 20% of AlgNa in its structure (H1, H2, and H4) present a greater number of groups capable of forming hydrogen bonding as compared to the samples with 10% of AlgNa in its composition. In the samples with 20% AlgNa, the chains interact with each other more. These chains are closer, and, therefore, the matrix becomes more compact, delaying the entry of fluid into the material and reducing the swelling, as can observed in Figure 4A.

#### 2.2.4. Kinetic Swelling Study

To have a more detailed idea of the build of the swelling process matrix, the experimental data were analyzed using Equation (2). The results are shown in Table 1. It can be seen that the value of *n* for all samples, both in swelling at pH 2 and PBS, is between 0.5 and 1. Therefore, it can be deduced that the swelling occurs by an anomalous process according to the model proposed. That is, in the swelling process, the diffusion phenomenon is simultaneously accompanied by the viscoelastic relaxation of the polymer chains (both processes have similar rates). In a previous work published by Goel et al. [15], which evaluated the influence of the inclusion of a cationic monomer on the swelling process of hydrogels of 2-hydroxyethylmethacrylate-co-[2-(methacryloyloxy)ethyl] trimethyl ammonium chloride, values of *n* greater than 0.5 were reported with the addition of small quantities of the cationic monomer, and, when increased, *n* values also increased, reaching values close to 1. According to the research conclusions, this occurs because the charged segments present in the hydrogel will repel each other to open the matrix and cause a more rapid diffusion of the water into the interior. In our work, this approach is not totally valid because the matrix contains AlgNa, which is an anionic polysaccharide whose -COO^−^ groups, as mentioned above, will be linked by electrostatic interactions with the N^+^(CH3)_3_ groups of the QCL, causing a contraction of the polymeric network that reduces the rate of diffusion of the surrounding fluid into the matrix. Therefore, a relationship between the concentration of QCL and the value of *n* will not be as proportional as can be seen in the results obtained by Goel et al.

### 2.3. Betamethasone Release Study

#### 2.3.1. FTIR Characterization of BTM Loaded Hydrogels

To identify the BTM in the hydrogels once charged, as well as to identify possible interactions between the matrix and the drug, a study was carried out using FTIR spectroscopy of the empty and loaded hydrogels. Figure 6 shows the FTIR spectrums of pure BTM, and three samples of AlgNa-g-poly(QCL-co-HEMA) hydrogels that are empty and loaded with BTM. In the FTIR spectra of pure BTM (Figure 6g), the characteristic absorption bands of this compound can be observed.

The signals observed at 1717 and 1658 cm^−1^ correspond to the C=O stretching vibration in the carbonyl group of the chain and the ketonic group of ring # 1 in the cortisol structure [39,40]. Additionally, bands at 1173 and 1086 cm^−1^ are attributed to asymmetric and symmetric stretching vibration of -C-O. The signals seen at 981 and 888 cm^−1^ can be attributed to P–O stretching vibration in phosphate moiety and the C–C skeleton vibration in the cyclopentane ring, respectively [40,41].

In the spectra of the loaded hydrogels, the characteristic bands of the BTM present in the spectrum of pure BTM are observed. As can be seen, there are no significant changes in the peaks, which signals that there is no demonstrable chemical interaction among the hydrogel networks and the groups present in the drug molecule. The drug is then trapped in the matrix only by physical interactions.

#### 2.3.2. Drug Release Study

One of the important parameters when designing a hydrogel-based, controlled delivery system for cartilage repair is its loading ability for the bioactive substance. The incorporation of the drug into the matrix can be performed by a swelling-diffusion process [42], as in our case. Drug retention can occur by physical or chemical immobilization in the polymeric network or through electrostatic interactions [43]. The drug load and loading efficiency of the studied hydrogels are presented in Table 2.

In previous works in which drug encapsulation studies have been carried out, the authors report that the swelling of the material [44,45], the concentration of the drug loading solution [46], and the interactions between the drug and the matrix [47] are the predominant factors on the loading of the drug into the matrix. The results observed in Table 2 show that the drug load in this work does not have a proportional relationship with the swelling degree, according to the results previously analysed in Section 2.2.3. However, it is observed that, when the concentration of HEMA increases in the formulation, the drug load also increases. In addition, for the same percentage of HEMA, the samples with 20% AlgNa have a higher drug load except for 10% of HEMA. Considering the described behavior and the molecular structure of BTM in which there are acceptor sites capable of forming hydrogen bonds (H-bond acceptor), such as the oxygens of the phosphate group located at the end of the molecule [48,49], it can be deduced that an interaction between the drug molecules and OH groups present in HEMA and AlgNa is taking place.

After immersion of the BTM-loaded hydrogels in a PBS solution, the amount of BTM released gradually increased over time, as can be seen in Figure 7. A burst was not observed in any of the samples, which suggests that the drug is homogeneously distributed in the matrix and is not collected only on the material surface. The release study was carried out for 8 h and it was observed that the samples H6 and H4 released all the encapsulated content before the fifth hour of study. The sample H5 release reached almost 100% during the studied time, while the rest of the samples (H2, H3, and H1) released 95%, 91%, and 83%, respectively.

The release processes are influenced by different factors, such as solubility of the encapsulated drug, crosslinking, and material composition, external factors such as pH or temperature, and possible interactions between the encapsulated drug and components of the polymeric matrix. According to the observed results, it can be deduced that, in our case, the swelling degree is not the fundamental variable that affects the drug release process since there is no direct relationship between the swelling degree previously determined and the release rate obtained for each sample. On the other hand, taking into account that the encapsulated BTM is a phosphate sodium salt and it is totally soluble in PBS, we can also rule out this factor as a variable determining the behavior of the release process. Then, evaluating the composition of the material, we can observe (as summarized in Table 2) that, when the concentration of HEMA in the matrix increases, the release of BTM decreases. For the same concentration of HEMA, when the concentration of AlgNa increases, the drug release decreases as well. This result is in accordance with the previous observation in the hydrogel loading process. In this case, the interaction between AlgNa and HEMA with the drug via H bonding causes the release of the drug to be delayed from the matrices that have a higher content of HEMA and AlgNa.

#### 2.3.3. Kinetic Drug Release Study

In order to know the kinetic mechanism of encapsulated BTM release, the in vitro experimental data were adjusted to the equation of the Korsmeyer-Peppas kinetic model, which is very useful when more than one mechanism is involved in the drug release or when accurate mechanisms are unknown [50]. As can be seen in Table 3, all the calculated *n* values are above 0.5 and the correlation coefficient with this model was high (R^2^ > 0.96). For samples H1 to H5, the value of *n* is between 0.5–1, indicating that the BTM released follows a non-Fickian diffusion mechanism, in which the drug released by diffusion and relaxation of the polymer chains occurs simultaneously. However, in the sample H6, the value of *n* is slightly higher than 1, which implies that the release is controlled by a case II diffusion, where the diffusion of the drug toward the outside of the matrix occurs very quickly when compared to the chain relaxation processes, which explains the accelerated release suffered by the drug in this matrix.

### 2.4. Cytocompatibility Studies

For the practical application of hydrogels in different biomedical fields, the developed material does not have inherent cytotoxicity. In vitro cytotoxicity standard ISO 10993-5 states that “reduction of cell viability by more than 30% is considered a cytotoxic effect”. In this study, the cytotoxicity of the blank hydrogels was investigated in C-28 and 3T3 cells and the results of the MTS assay are presented in Figure 8A,B respectively.

The results show that the viability of the hydrogel extract-treated cells was higher than 90% after 24 h and even higher than 100% after 48 and 72 h of study in samples H1 to H5. The cell viability values obtained indicate that these materials are non-cytotoxic and also favor proliferation against the cells studied. Conversely, sample H6 shows high cytotoxicity for both cells evaluated. The number of viable cells are less than 20% in both kinds of cell cultures when they interact with the hydrogel extract in all evaluated times. This could be conditioned by the previous result obtained in the FTIR analysis, where the signal corresponding to the vinylic group is observed, indicating that an unreacted monomer is present in the hydrogel and known monomers are molecules with a high toxicity.

After quantitative analysis results were obtained by the MTS assay, a qualitative analysis was carried out via the Live/Dead^®^ assay to visualize the distribution of living and dead cells after 7 days of being seeded in different samples of AlgNa-g-poly(QCL-co-HEMA) hydrogels. Figure 9A,B show the microphotographs of cells seeded on each hydrogel after the staining. In the microphotographs obtained for the samples from H1 to H5 (Figure 9A(a–e),B(a–e)), most cells observed are alive. Additionally, cells are spread on the hydrogel’s surfaces and elongated morphologies are also observed, which is a sign that cells are healthy. In some images, it is possible to observe grouped cells forming colonies, which also shows that the environment where they are seeded is favorable for their proliferation. On the contrary, in sample H6, the number of dead cells is higher than the number of living cells, which is in agreement with the result obtained by the MTS assay discussed above.

According to the results obtained, it can be concluded that the AlgNa-g-poly(QCL-co-HEMA) hydrogels, of samples from H1 to H5, are biocompatible against C28 and 3T3 cells, demonstrating their potential as a scaffold for cartilage tissue engineering and a drug delivery system.

## 3. Materials and Methods

### 3.1. Materials

Commercial 2-hydroxyethylmethacrylate (HEMA, Sigma-Aldrich, Amsterdam, The Netherlands), 2-acryloxyethyl-trimethylammonium chloride (QCL, Sigma-Aldrich, Amsterdam, The Netherlands), potassium persulfate (K_2_S_2_O_8_, Honeywell Fluka, Fisher Scientific, Madrid, Spain), and N, N methylenebisacrylamide (MBA, Merck BV, Amsterdam, The Netherlands) were used as purchased. Sodium alginate from *Macrocystis periferia* (viscosity-average molecular weight of 3.83 × 10^4^ g/mol) was purchased from Sigma-Aldrich Chemie GmbH, Taufkirchen, Germany. Dulbecco’s phosphate buffered saline (PBS), Dulbecco’s Modified Eagles Medium (DMEM, high glucose, with Glutamax^TM^), fetal bovine serum (FBS), penicillin, and streptomycin were purchased from Life Technologies (Breda, The Netherlands). 3-(4,5-dimethylthiazol-2-yl)-5-(3-carboxymethoxyphenyl)-2-(4-sulfophenyl)-2H-tetrazolium (MTS, Promega Benelux BV, Leiden, The Netherlands), Calcein-AM/ethidium homodimer-1 Live/Dead^®^ assay kit (Invitrogen, Carlsbad, CA, USA) were used for cell studies.

### 3.2. Preparation of AlgNa-g-Poly(QCL-co-HEMA) Hydrogels

AlgNa-g-poly(QCL-co-HEMA) hydrogels were prepared by graft copolymerization of QCL and HEMA onto the sodium alginate (AlgNa) chain in the presence of potassium persulfate (1% wt) as a thermal initiator, and N, N methylenebisacrylamide (MBA, 2% wt) as a crosslinking agent. The composition of the hydrogels is listed in Table 4. The final volume of the mixture used was 10 mL. The calculated amount of AlgNa was dissolved in 5 mL of distilled water. KPS and MBA were added under constant stirring. Then, the monomer quantity was determined for each formulation. The final solution was placed in a glass tube, degassed for 30 min in an N_2_ atmosphere to eliminate the dissolved oxygen in the system, and sealed under vacuum. The glass tubes were placed in a thermostatic water bath at 60 °C for 12 h. The hydrogels obtained were cut into discs for later studies.

### 3.3. Hydrogel Characterizations

#### 3.3.1. Instrumental Analysis

Dried hydrogel samples were characterized by Fourier transform-infrared (FT-IR) spectroscopy and scanning electron microscopy (SEM). The FT-IR spectra were obtained on a Shimadzu IRSpirit-T spectrophotometer (Shimadzu Co., Kyoto, Japan), equipped with an attenuated total reflectance accessory (ATR). The absorption spectra were recorded in the spectral range of 4000–600 cm^−1^ with a resolution of 4 cm^−1^ for 32 scans. Scanning electron microscopic observations were performed to visualize pores and morphology of the hydrogels using a SEM-JEOL JSM-6360LV microscope (Jeol Co., Tokyo, Japan). Small pieces of hydrogels were sectioned and freeze-dried to observe the internal structure. The samples were placed on an aluminum mount, sputtered with gold palladium, and then scanned at an accelerating voltage of 25 kV.

#### 3.3.2. Swelling Studies

The swelling kinetics of the terpolymeric hydrogels were studied in different media (HCl, pH 2.0 and PBS, pH 7.4). The samples previously weighed were immersed in 20 mL of each swelling medium at 37 °C in a thermostatic bath for 24 h to reach the swelling equilibrium. At established time intervals, the hydrogels were carefully taken out from the solution. The excess of water on the surface was removed with a filter paper, and then weighted. In all the experiments, the water uptake was determined by gravimetric measurements using an analytical balance. The swelling degree was calculated as follows.
(1)$$swelling\;degree\;\left(\%\right)=\frac{w_{s}-w_{d}}{w_{d}}\times 100$$
where *w_s_* and *w_d_* are the weights of the swollen and dry hydrogels, respectively. Experiments were carried out in triplicate.

In order to determine the mechanism of diffusion of the pH 2.0 solution and PBS into the hydrogels, the following equation was used.
(2)$$\frac{w_{t}}{w_{\infty }}=Kt^{n}$$
where *K* is a kinetic constant characteristic of the polymer system, *w_t_* is the mass of solvent absorbed at time *t*, *w*_∞_ is the mass of solvent absorbed at equilibrium, and *n* is an empirical number called a transport exponent. Equation (2) is applied to the initial stages of swelling up to 60%. A value of *n* = 0.5 is taken as an indication that the process is diffusion controlled (Fickian diffusion) or Case I transport, whereas, for *n* = 1.0, the swelling is considered to be controlled by the chains relaxation (Case II transport). When the value of *n* lies between 0.5 and 1.0, the process is considered anomalous (non-Fickian diffusion, which is a combination of diffusion and relaxation) [15].

### 3.4. Betamethasone In Vitro Release Study

To study the behavior of the developed hydrogels as drug delivery systems, betamethasone as a model drug was loaded into them. The dry hydrogels (pieces of 40–70 mg) were immersed in 10 mL of 1 mg/mL betamethasone solution for 24 h. After incubation, the BTM-loaded hydrogels were taken out, rinsed with distilled water to remove the excess drug on the surface, and dried at room temperature for 3 days. The loaded hydrogels were characterized by FTIR spectroscopy in order to observe if they suffered any changes in the structure after incubation in BTM solution or if any chemical interaction between the drug and matrix occurred.

The amount of drug loaded (DL) in the hydrogel discs (ug of drug/mg of hydrogel) and the loading efficiency (LE) were calculated by the following equations [46,51].
(3)$$DL\;\left(\mu g\;of\;drug/mg\;of\;hydrogel\right)=\frac{C_{1}V_{1}-C_{2}V_{2}}{m_{H}}$$
(4)$$LE\;\left(\%\right)=\frac{C_{1}V_{1}-C_{2}V_{2}}{C_{1}V_{1}}\times 100$$
where *C*_1_ is the initial concentration of the loading BTM solution (mg/mL), *V*_1_ is the initial volume of the BTM solution (mL), *C*_2_ and *V*_2_ are the final concentration and remaining volume of the BTM solution after the hydrogel’s incubation including 5 mL of water used to rinse the loaded hydrogels. *m_H_* is the weight of the hydrogel disc before being loaded. The amount of BTM was determined spectrophotometrically at 242 nm using a BTM standard calibration curve prepared at concentrations between 5 and 40 µg/mL.

The in vitro release studies were performed in PBS at pH 7.4 and 37 °C. Briefly, the loaded hydrogel samples were incubated in 5 mL of PBS and, at selected time intervals, 0.5 mL of medium was extracted and replaced with an equal volume of fresh PBS. The BTM concentration released was obtained by the same method mentioned above and the cumulative released amount (% BTM released) in each time interval was calculated according to Equation (5).
(5)$$\%BTM_{released}=\frac{m_{r}}{m_{t}}*100$$
where *m_r_* is the released amount of BTM at selected time intervals and *m_t_* is the initial amount of BTM loaded in the dry hydrogels. Experiments were run in triplicate.

The release mechanism from AlgNa-g-poly(QCL-co-HEMA) hydrogels was investigated using released experimental data and the Korsmeyer-Peppas power law equation (Equation (2)) [15]. In this case, *w_t_* is the mass released at time t and *w*_∞_ is the total mass of BTM encapsulated in the matrix.

### 3.5. Cell Culture and Cytotoxicity Assay

Human chondrocyte C-28 and fibroblast 3T3 cells were cultured in DMEM containing 10% fetal bovine serum (FBS) and 1% antibiotics (penicillin-streptomycin) at 37 °C under a 5% CO_2_ humidified atmosphere.

The in vitro cytotoxicity of the hydrogels was evaluated based on their impact on cell structures such as mitochondria by the MTS assay (3-(4,5-dimethylthiazol-2-yl)-5-(3-carboxymethoxyphenyl)-2-(4-sulfophenyl)-2H-tetrazolium). The indirect extraction method to evaluate hydrogel samples’ cytotoxicity was used according to the ISO 10993-5 standard and previous literature [25,52]. The hydrogel extract solutions were prepared by incubating samples previously sterilized, in DMEM culture medium for 48 h at 37 °C. After incubation, the extract solutions were filtered through a 0.22-µm syringe filter. Alternatively, C-28 and 3T3 cells were seeded in 96-well plates at a density of 1 × 10^4^ cells per well and cultured in 5% CO_2_ at 37 °C for 24 h. Afterward, the medium in each well was replaced with 100 µL of the hydrogels’ extract solutions and the treated cells were incubated for 24, 48, and 72 h. After the incubation time, the medium was discarded, the MTS solution was added, and plates were incubated for another 3 h in darkness. The absorbance of each well was measured by a micro-plate reader (VersaMax, Molecular Devices, San José, CA, USA, Program Softmax Pro) at 490 nm. Cells incubated only with DMEM culture medium were used as a negative control (100%), and the relative cell viability of the treated groups was calculated according to the next equation.
(6)$$\%\;Cell\;viability=\frac{OD_{samples}}{OD_{control}}\times 100$$

In order to evaluate the biocompatibility of materials after 7 days in contact with C28 and 3T3 cells, the Live/Dead^®^ assay was made. Live/Dead^®^ is a quick and easy two-color assay where Calcein AM fluoresces green upon the reaction of intracellular esterase and stains live cells. Ethidium homodimer-1, which binds to the DNA of dead membrane compromised cells, stains dead cells (red). First, small 2–3 mm thin disks of hydrogels were placed in 48-well plates overnight in DMEM medium. Then, the medium was removed, an aliquot of C28 human chondrocytes or C3T3 fibroblast suspension (2 × 10^5^ cells) was seeded on the surface of the hydrogel disk and incubated for 6 h at 5% CO_2_ and 37 °C to allow cell attachment. Subsequently, cell-seeded hydrogels were transferred into new 24-well plates. Then, 1 mL of DMEM media was added on each well and incubated for 7 days. After every 24 h, culture medium was replaced. At 7 days, the hydrogels were rinsed with PBS and stained with Calcein AM/ethidium homodimer-1 using the Live-Dead Assay Kit (Invitrogen), according to the manufacturers’ instructions. The images of the C28 and 3T3 cells on the surface hydrogels were captured by a fluorescence microscope (Leica DM 5500 B, Leica Microsystems GmbH, Wetzlar, Germany).

### 3.6. Statistical Analysis

Graphs and statistics were performed with OriginPro 2021 (OriginLab Corp., Northampton, MA, USA). Data are reported as mean ± standard deviation (SD), unless stated otherwise. Error bars represent the SD calculated from tests of triplicate measurements for each scaffold. Statistical analysis was significant by a One-Way Analysis of Variance (ANOVA) for *p* < 0.05 or *p* < 0.01, according to the *t*-test for two samples or a multiple samples’ comparison.

## 4. Conclusions

In this study, a series of AlgNa-g-poly(QCL-co-HEMA) hydrogels were successfully synthesized by graft co-polymerization of QCL and HEMA onto the sodium alginate chain. Through FTIR characterization, it was possible to confirm the formation of a graft co-polymer and the structural composition of these materials, constituted mainly by groups such as -OH, -COOH, -C=O and N^+^(CH_3_)_3_. The morphological study shows that an internal structure is porous in all samples, which is an advantage in materials for biomedical applications. The swelling study showed that the pH of the medium and the composition of the hydrogels affect the swelling degree, and, based on the applied kinetic model, it was observed that the fluid penetrates the polymer network through an anomalous process. In vitro release of BTM loaded into the hydrogels were carried out for 8 h and the studies showed an effective BTM release from hydrogels in a controlled manner conditioned by a simultaneous process of diffusion and relaxation of the polymeric chains, according to the Korsmeyer-Peppas kinetic model used in the analysis. In the case of in vitro cytocompatibility studies, it was confirmed through MTS and Live/Dead^®^ assays that samples from H1 to H5 are cytocompatible and allow the adhesion and cell proliferation against C28 and 3T3 cells, while, in the sample H6, the cell viability was less than 20%, which demonstrates that a QCL concentration higher than 70% is not suitable for biomedical materials. We can conclude that AlgNa-g-poly(QCL-co-HEMA) hydrogel matrices obtained (compositions from H1 to H5) are promising materials as platforms for chondrocyte proliferation and drug delivery.

## Figures and Tables

**Figure 1 ijms-22-05730-f001:**
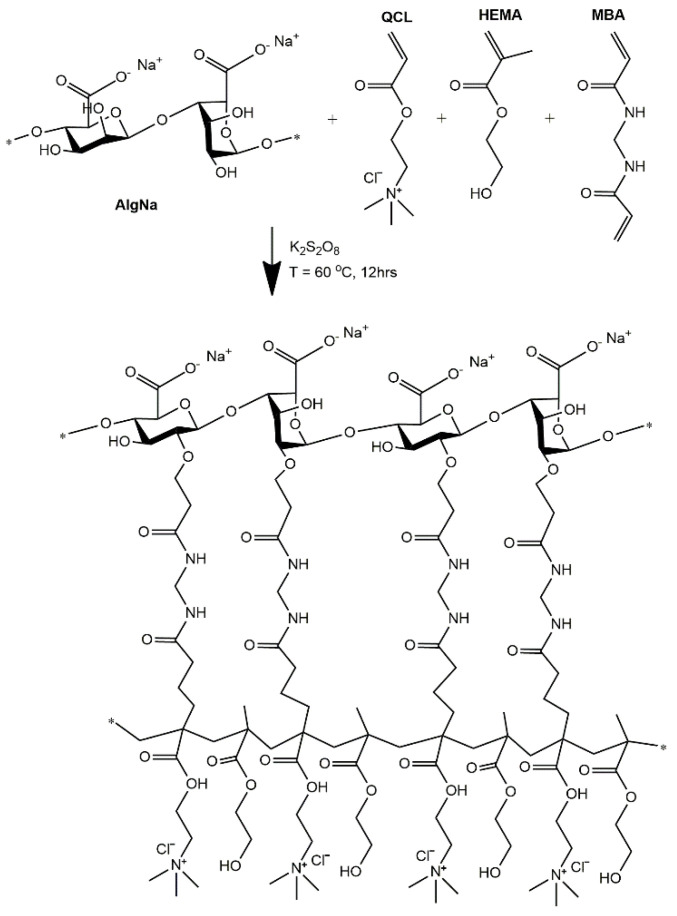
Schematic representation of chemical structures of polymer, monomers, and hypothetically cross-linked structures of AlgNa-g-poly(QCL-co-HEMA) hydrogels.

**Figure 2 ijms-22-05730-f002:**
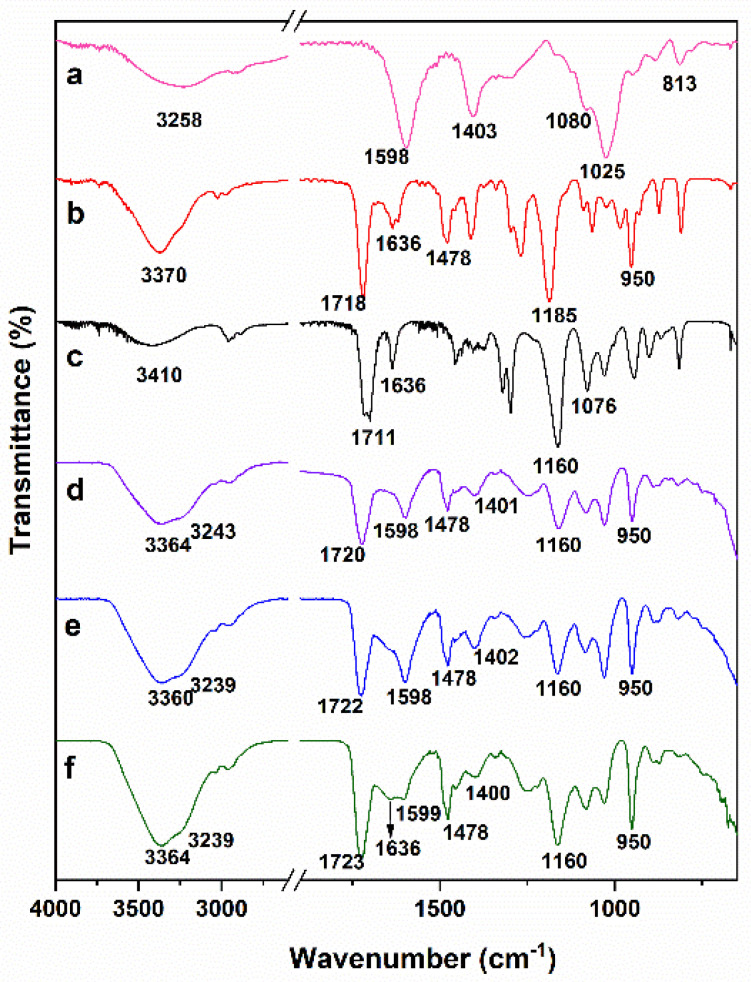
FTIR spectra of AlgNa (**a**), QCL (**b**), HEMA (**c**), hydrogel H2 (**d**), hydrogel H4 (**e**), and hydrogel H6 (**f**).

**Figure 3 ijms-22-05730-f003:**
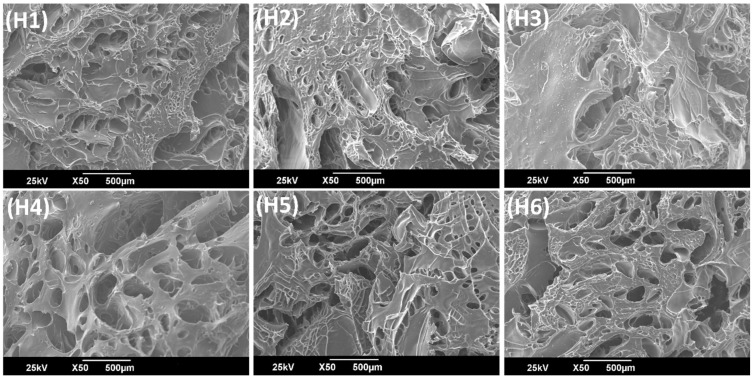
Scanning electron microscopy images of AlgNa-g-poly(QCL-co-HEMA) hydrogels.

**Figure 4 ijms-22-05730-f004:**
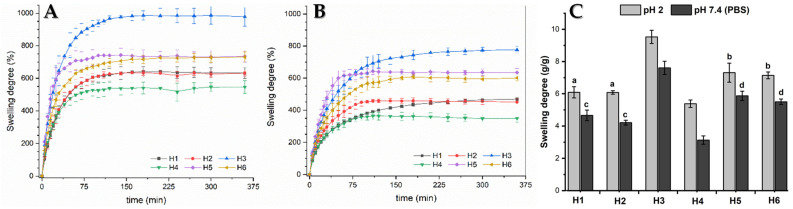
Swelling studies. (**A**) and (**B**) hydrogels’ profiles at pH 2.0 and 7.4, respectively. (**C**) Maximum swelling at equilibrium in more detail. There are no significant differences between two compared samples at same pH, for a and b (*p* ≥ 0.05), or for c and d (*p* ≥ 0.01).

**Figure 5 ijms-22-05730-f005:**
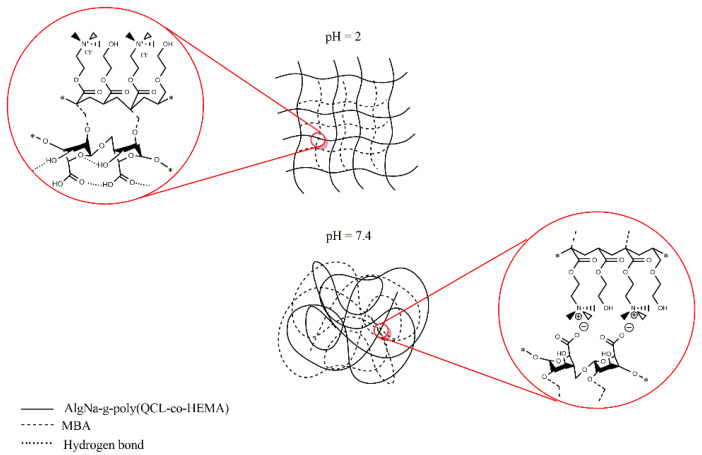
Schematic representation of the interactions between functional groups in the polymeric network chains at pH 2.0 and pH 7.4.

**Figure 6 ijms-22-05730-f006:**
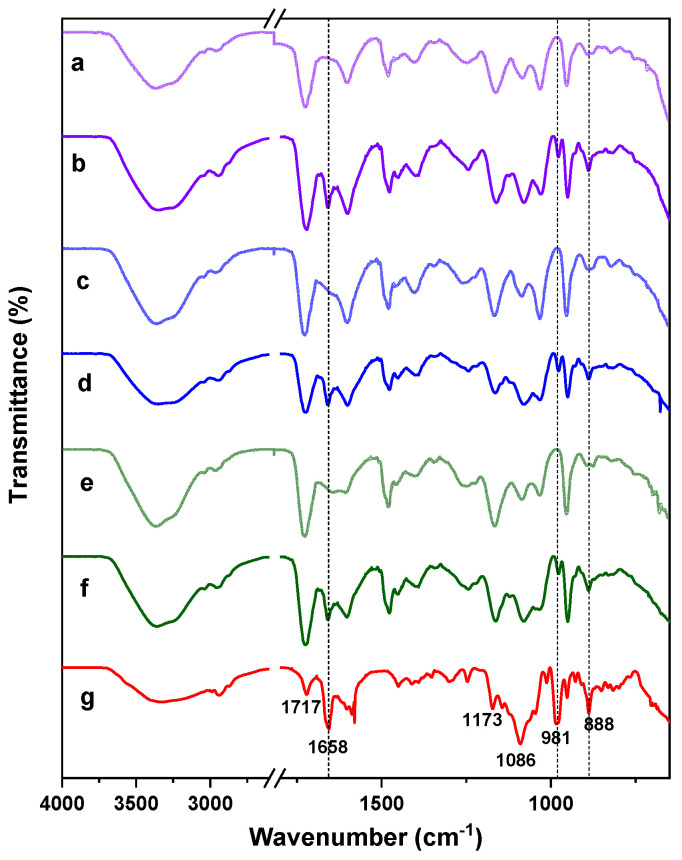
FTIR spectra of BTM (**g**), hydrogel H2 empty and loaded (**a**,**b**), hydrogel H4 empty and loaded (**c**,**d**), and hydrogel H6 empty and loaded (**e**,**f**).

**Figure 7 ijms-22-05730-f007:**
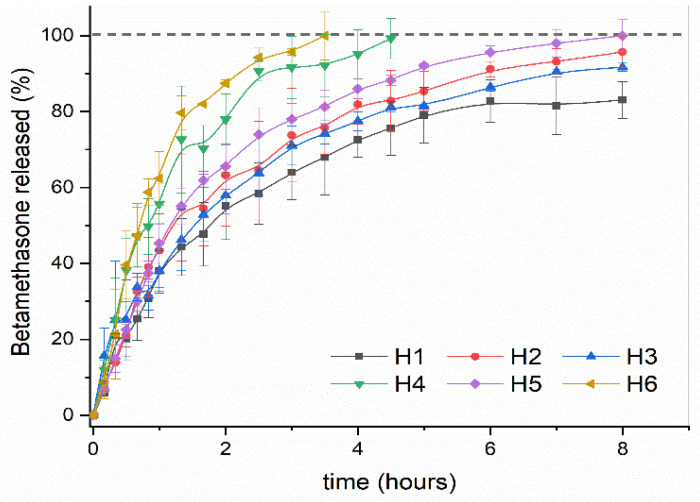
Cumulative in vitro release study of Betamethasone from AlgNa-g-poly(QCL-co-HEMA) hydrogels in PBS, pH = 7.4.

**Figure 8 ijms-22-05730-f008:**
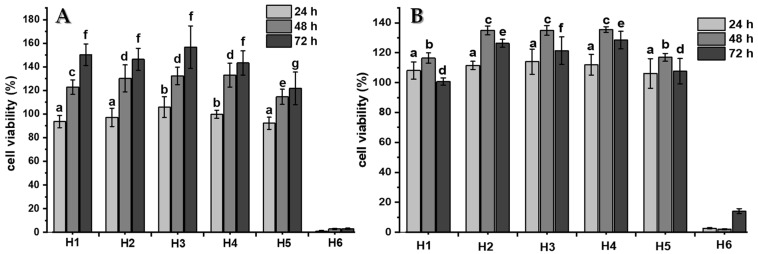
Cytotoxicity evaluation of the hydrogel’s extracts exposed to C28 (**A**) and 3T3 (**B**) cells using the MTS assay method. There are no significant differences among several compared samples at the same time interval, for any letters, in each graph (*p* ≥ 0.05).

**Figure 9 ijms-22-05730-f009:**
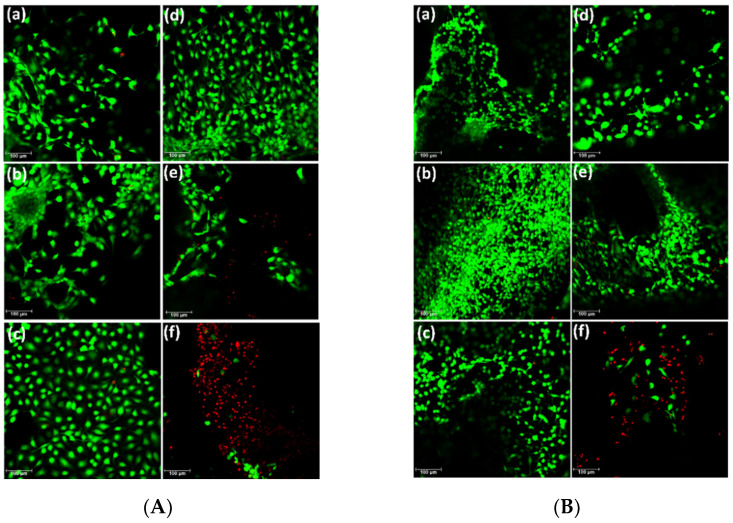
(**A**) Microphotographs of C28 cells seeded on AlgNa-g-poly(QCL-co-HEMA) hydrogels after 7 days, (**a**) H1, (**b**) H2, (**c**) H3, (**d**) H4, (**e**) H5, (**f**) H6. (**B**) Microphotographs of 3T3 cells seeded on AlgNa-g-poly(QCL-co-HEMA) hydrogels after 7 days, (**a**) H1, (**b**) H2, (**c**) H3, (**d**) H4, (**e**) H5, (**f**) H6.

**Table 1 ijms-22-05730-t001:** Swelling kinetic constant (k), transport exponent (*n*), and coefficient of determination (R^2^), according to Equation (2).

Samples	pH = 2.0			PBS, pH = 7.4		
	k	*n*	R^2^	k	*n*	R^2^
H1	38 ± 3	0.55 ± 0.02	99.67	33 ± 2	0.74 ± 0.02	99.90
H2	40 ± 4	0.60 ± 0.03	99.56	41 ± 3	0.69 ± 0.03	99.79
H3	52 ± 4	0.60 ± 0.03	99.55	65 ± 3	0.69 ± 0.02	99.93
H4	35 ± 1	0.59 ± 0.01	99.97	47 ± 2	0.64 ± 0.01	99.95
H5	45 ± 4	0.71 ± 0.03	99.80	68 ± 6	0.72 ± 0.03	99.84
H6	49 ± 3	0.59 ± 0.02	99.72	53 ± 2	0.69 ± 0.02	99.95

**Table 2 ijms-22-05730-t002:** Relation between hydrogels’ composition and drug load, loading efficiency, and BTM released.

Samples	Composition (%)			Drug Load (µg of Drug/mg of Hydrogel)	Loading Efficiency (%)	BTM Released (%)
	HEMA	QCL	AlgNa			
H6	10	80	10	58.0	31.8	100 *
H4	10	70	20	57.8	27.9	100 **
H5	20	70	10	74.1	52.7	99.7
H2	20	60	20	99.8	37.6	95.5
H3	30	60	10	113.7	58.8	91.4
H1	30	50	20	137.4	59.6	83.2

* BTM released before 4 h. ** BTM released before 5 h.

**Table 3 ijms-22-05730-t003:** Drug release rate constant (k), diffusion exponent (*n*), and coefficient of determination (R^2^), according to the Korsmeyer-Peppas model.

Samples	Korsmeyer-Peppas		
	k	*n*	R^2^
H1	2.7 ± 0.6	0.63 ± 0.05	97.53%
H2	1.8 ± 0.6	0.76 ± 0.08	96.24%
H3	4.4 ± 0.7	0.54 ± 0.04	98.28%
H4	2.9 ± 0.9	0.73 ± 0.08	97.82%
H5	1.0 ± 0.1	0.93 ± 0.04	99.55%
H6	1.0 ± 0.4	1.0 ± 0.1	98.57%

**Table 4 ijms-22-05730-t004:** Composition in a reaction mix of different hydrogels (total mass 2 g).

Sample	QCL (% wt)	HEMA (% wt)	AlgNa (% wt)
H1	50	30	20
H2	60	20	20
H3	60	30	10
H4	70	10	20
H5	70	20	10
H6	80	10	10

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
