# Peer review of "Synthesis and Evaluation of AlgNa-g-Poly(QCL-co-HEMA) Hydrogels as Platform for Chondrocyte Proliferation and Controlled Release of Betamethasone"

_ijms, 2021, doi:10.3390/ijms22115730_

Round 1

Reviewer 1 Report

The manuscript describes the synthesis and characterisation of a graft copolymers based on the biocompatible natural polymer alginate.

The manuscript required an in depth proof reading as it contains numerous grammatical and spelling errors.

The type of study performed does not support the claim that these materials are good for cartilage tissue engineering so the title of the paper could be revised to say refer more in general to biomedical applications and drug delivery, consequently the introduction would need changing as well.

Lines 63 and 64: check use of symbols for alpha and beta

Line 71: alginate does not have any specific site for cell attachment

Line 94: cite the similar reports mentioned.

Lines 101-102: rephrase as it is not clear.

Figure1: specify that this is a hypothetical structure as many different side reactions might be occurring and several different polymers are surely produced in the process.

Figure 4: please report the statistical analysis of graph 4C, a t test to show difference between 2 pH values tested and one way Anova to show differences between different polymers should be reported.

Lines 202-203: please explain better why having more alginate would reduce the swelling

Section 2.3.2: before reporting the drug release it would be important to analyse the data of drug loading. Percent entrapment and loading efficacy should be reported, analysed and discussed. The release mechanism is discussed as solely dependent on H-bond interactions between the hydrogel and the drug, however the hydrogel has positive charges and the drug presents negative charges therefore consideration of ionic interactions should also be made. Are these completely excluded on the basis of the FTIR results?

Figure 8: please report the statistical analysis of the data

Line 338: please report more details about the alginate used

Line 400: report the concentration range of betamethasone for which you obtained a linear correlation when constructing the calibration curve.

Line 403: does V2 also include the volume of water used to wash the gels?

Line 457: even though it is mentioned that statistical analysis has been performed the results are not reported in the manuscript

Reviewer 2 Report

The submitted paper tackles an interesting topic, but your manuscript has some significant flaws. I hope that the attached list of observations will help you in your research. I wish you the best of luck!

Round 2

Reviewer 1 Report

I wish to thank the authors for responding to comments in a detailed manner.

Most comments have been addressed satisfactorily my only remaining concern is the description of the statistical analysis as it is not clear, also the way in which the statistical data have been presented in the figures is unclear.

Further proof reading and editing of English is also required. 

Reviewer 2 Report

Congratulations for your work. Below you may find three minor observations.

Line 280: “Drug retention can be occurred by physical or chemical immobilization in the polymeric network or through electrostatic interactions [43]. The drug load and loading efficiency of the hydrogels studied show in Table 2.”, I suggest rephrasing to “Drug retention can occurred by physical or chemical immobilization in the polymeric network or through electrostatic interactions [43]. The drug load and loading efficiency of the studied hydrogels is presented in Table 2.”

Line 295: the phrase “In addition, for the same % of HEMA, the samples with 20% AlgNa have a higher drug load” is not true for all compositions - H6: 10% HEMA, 80% QCL, 10% AlgNa - drug load 58 um/mg compared to H6: 10% HEMA, 70% QCL, 20% AlgNa - drug load 57.8 um/mg. - Please rephrase.

Line 296: “Considering the describe behavior and the molecular structure…” I suggest modifying to “Considering the described behavior and the molecular structure…”
